# Reinforcement Learning and Adaptive Sampling
# for Optimized DNN Compilation

**Byung Hoon Ahn** [1]  **Prannoy Pilligundla** [1]  **Hadi Esmaeilzadeh** [1]

## Abstract

Achieving faster execution with shorter compilation time can enable further diversity and innovation in neural networks. However, the current paradigm of executing neural networks either relies on hand-optimized libraries, traditional compilation heuristics, or very recently, simulated annealing and genetic algorithms. Our work takes a unique approach by formulating compiler optimizations for neural networks as a reinforcement learning problem, whose solution takes fewer steps to converge. This solution, dubbed RELEASE, comes with a sampling algorithm that leverages clustering to focus the costly samples (real hardware measurements) on representative points, subsuming an entire subspace. Our adaptive sampling not only reduces the number of samples, but also improves the quality of samples for better exploration in shorter time. As such, experimentation with real hardware shows that reinforcement learning with adaptive sampling provides 4.45×speed up in optimization time over AutoTVM (Chen et al., 2018b), while also improving inference time of the modern deep networks by 5.6%. Further experiments also confirm that our adaptive sampling can even improve AutoTVM's simulated annealing by 4.00×.

## 1. Introduction

Deep neural networks (DNNs) have pushed the boundaries in image classification (Krizhevsky et al., 2012; Sermanet et al., 2013; Simonyan & Zisserman, 2014; He et al., 2016; Szegedy et al., 2015; Howard et al., 2017), automatic speech recognition (Mohamed et al., 2011; Graves et al., 2013;

Amodei et al., 2016; Miao et al., 2015), autonomous decision making (Mnih et al., 2015; Silver et al., 2016; Mnih et al., 2016; Levine et al., 2016; Lenz et al., 2015; Mirhoseini et al., 2017), etc. The enormous computational intensity of DNNs have resulted in developing either hand-optimized kernels, such as NVIDIA cuDNN (Chetlur et al., 2014) or Intel MKL (MKL, 2009) that serve as backend for a variety of programming environment such as (Abadi et al., 2015; Jia et al., 2014; Paszke et al., 2017; Chen et al., 2015; Team et al., 2016). However, the complexity of the tensor operations in DNNs and the volatility of algorithms, which has led to unprecedented rate of innovation (LeCun, 2019), calls for developing automated compilation frameworks. To imitate or even surpass the success of hand-optimized libraries, recent research has developed stochastic optimization passes for general code, STOKE (Schkufza et al., 2013), and neural network code, AutoTVM (Chen et al., 2018b) and TensorComprehensions (Vasilache et al., 2018). AutoTVM uses simulated annealing and STOKE and TensorComprehensions rely on genetic algorithms to search the space of optimized code for neural networks. AutoTVM takes a further inspiring step and leverage boosted trees (Chen & Guestrin, 2016) as part of the search cost model to avoid measuring the fitness of each solution (optimized candidate neural network code), and instead predict its fitness. Even with these innovations the optimizing compilation time can be around 10 hours for ResNet-18 (He et al., 2016).

As such, this paper sets out to significantly reduce the compilation time and offer automation while avoiding dependence on hand-optimization, potentially enabling far more diverse tensor operations in next generation neural networks. We tackle this challenge from two fronts and makes the following contributions:

(1) Formulating optimizing compilation of neural networks as a *Reinforcement Learning (RL)* problem in contrast to simulated annealing and genetic algorithms of prior works, as a result requiring fewer steps to converge to even better or same quality solution.

(2) Devising an *Adaptive Sampling* algorithm that leverages clustering to focus on representative samples from different subspaces of possible solutions (optimized code), reducing the number of costly hardware measurements while maintaining high relevance to the search.

[1]Department of Computer Science and Engineering, University of California, San Diego, California, USA. Correspondence to: Byung Hoon Ahn <bhahn@eng.ucsd.edu>, Prannoy Pilligundla <ppilligu@eng.ucsd.edu>, Hadi Esmaeilzadeh <hadi@eng.ucsd.edu>.

*Reinforcement Learning for Real Life (RL4RealLife) Workshop in the $36^{th}$ International Conference on Machine Learning*, Long Beach, California, USA, 2019. Copyright 2019 by the author(s).

Real hardware experimentation with modern DNNs (AlexNet, VGG-16, and ResNet-18) on a high-end GPU (NVIDIA Titan Xp), shows that the combination of these two innovations, dubbed RELEASE, yields 4.45× over the leading framework, AutoTVM, that even aims to minimize compilation time with innovative cost models. RELEASE is publicly available as open-source at https://bitbucket.org/act-lab/release.

## 2. Optimizing Compilation for DNNs

### 2.1. Compilation Workflow

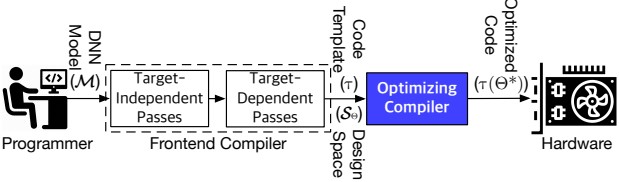

*Figure 1.* Overview of our model compilation workflow. Scope of this work is the optimizing compiler in the above diagram.

Figure 1 illustrates how a compiler for neural networks takes a DNN ($\mathcal{M}$) and emits an optimized code ($\tau(\Theta^*)$) that runs the model efficiently. This flow is commensurate with TensorComprehesions (Vasilache et al., 2018) and TVM (Chen et al., 2018a), using which we implemented the RELEASE optimizing compiler that will also be released as a separate package for adoption in other frameworks. The first phase of the workflow is the frontend compiler which performs the translation from the compiler and applies target-independent and white-box target-dependent optimizations that do not incorporate a measure of runtime. The next stage is a black-box optimization pass, called optimizing compiler, that given a measure of performance at runtime from the hardware can further optimize the code. RELEASE falls in this class by offering a RL-based optimizing compiler that also comes with an adaptive sampling algorithm.

Target-independent passes transform the input DNN model without specificity to the target hardware. Operator fusion and data layout transformation in TVM (Chen et al., 2018a) are some examples of these passes, which lie in the same category as dead-code elimination or loop-invariant code motion in LLVM (Lattner & Adve, 2004). Target-dependent passes, on the other hand, the compiler takes the hardware architecture (target) into account while optimizing the program, but does not actively leverage runtime measures.

### 2.2. Optimizing Compiler for Neural Networks

*Optimizing Compilers* (Kennedy & Allen, 2001), utilize runtime information, to further optimize the code. RELEASE, STOKE (Schkufza et al., 2013), AutoTVM (Chen et al., 2018b), the autotuner in TensorComprehensions (Vasilache et al., 2018) as well as profile-driven passes (Chang et al., 1991; Novillo, 2014) fall in this category. Optimizing compilers usually take a black-box approach and use hardware

*Table 1.* Example of knobs constituting the dimensions of the design space while optimizing convolution kernels.

| DIMENSION | DETAILS |
|---|---|
| tile_f, tile_y, tile_x | Tiling and binding # of filters height, width of feature maps. |
| tile_rc, tile_ry, tile_rx | Tiling and binding # for reduction axis such as channels, height, width of filters. |
| auto_unroll_max_step | Threshold of # of steps in the loop to be automatically unrolled in the CodeGen phase. |
| unroll_explicit | Explicit hint for CodeGen phase to unroll loop. |

measurements to configure the optimization based on a measure of fitness ($f$) for each solution. Optimizing compilers for neural networks make this problem more tractable by restricting the output code to a set of configurable templates ($\tau$) with tunable knobs ($\theta$). An optimizing compiler for neural networks can be formulated as:

$$\Theta^* = \underset{\Theta}{\text{argmax}}\, f(\tau(\Theta)), \qquad \text{for } \Theta \in \mathcal{S}_\Theta. \qquad (1)$$

A combinations of assignment to the knobs is said to be a configuration ($\Theta = (\theta_1, \theta_2, ..., \theta_n)$) while the dimensions of the design space ($\mathcal{S}_\Theta$) is defined by the knobs. As such, in (1), an optimizing compiler starts from a code template ($\tau$) for each layer, and makes use of a search algorithm and real hardware measurements to efficiently find the best configuration ($\Theta^*$) within the design space defined by the knobs. In this context, there are three variables that determine the effectiveness of the optimizing compiler: (1) a large and diverse enough design space (knobs) that covers a variety of transformations, (2) an effective search algorithm to adequately navigate this space, and (3) a mechanism to cut down the number of costly hardware measurements that check the fitness of a solution. Table 1 shows the search space for performing convolution on a GPU. In GPUs, it is crucial that the code (1) maximizes data reuse, (2) uses the shared memory wisely, and (3) minimizes bank conflicts. The knobs optimize various aspects of the execution, including tiling (e.g., tile_x, tile_y, ...), unrolling (e.g., auto_unroll_max_step and unroll_explicit). These knobs define a search space with $10^{10}$ possibilities. Given that vastness of the search space, the challenge is designing an effective search algorithm and a mechanism that reduces the cost of each step in the search (i.e. reducing the need to measure the hardware).

## 3. Challenges and Design Objectives

### 3.1. Challenges

Even with the advances from prior works (Chen et al., 2018a; Vasilache et al., 2018; Chen et al., 2018b), optimizing compilation can be around 10 hours for ResNet-18 (He et al.,

2016) with 12 convolution layers. This long optimization time gets more prominent in deeper or wider networks with models with more larger layers to optimize. Such long optimization time results from naive stochastic search of simulated annealing or genetic algorithm (Davis, 1987) and excessive number of real hardware measurements from simple sampling. Therefore, having large and diverse enough design space provided a priori, variables that determine the effectiveness of the optimizing compiler can be narrowed down to two subproblems: (1) developing an efficient search algorithm, and (2) reducing the number of times the compiler reaches for real hardware measurements.

### 3.2. Design Objectives

**Improving efficacy of search algorithm.** One strategy to approach this problem is to do a brute force search. However, when the design space could be as large as $10^{10}$, (brute force) optimization becomes too time-consuming leaving it unrealistic or it fails to provide a reasonable solution making it unpractical a solution. Another strategy is to incorporate random search (Chen et al., 2018a) or bio-inspired meta-heuristic like genetic algorithms (Vasilache et al., 2018; Chen et al., 2018a; Ragan-Kelley et al., 2017; Ballal et al., 2015; Cooper et al., 1999; Ansel et al., 2009) to enhance efficiency of the search. Prior works (Chen et al., 2018b; Shen, 2009; Mei et al., 2002) have also used simulated annealing in the context of compiler optimization problem because it statistically guarantees finding an optimal solution given an energy function.

Although previous work (Chen et al., 2018b) finds reasonable configurations with the interplay of simulated annealing and cost models with boosted trees (Chen & Guestrin, 2016), simulated annealing is known for its slow speed and could be an overkill. Furthermore, simulated annealing is oblivious to the gradual changes in the cost model and naively trusts the estimation. This leads simulated annealing based search doing redundant work during the search, as a result leaving room for improvement in the effectiveness and efficiency of the search. This calls for a more intelligent search algorithm ($\mathcal{A}^*$) that meets following objectives:

$$s_\Theta^* = \operatorname*{argmax}_{s_\Theta \subset \mathcal{S}_\Theta} \left( P(f_{ideal}(\tau) - \max_{\Theta \in s_\Theta} f(\tau(\Theta)) = 0) \right) \quad (2)$$

$$\mathcal{A}^* = \operatorname*{argmin}_{\mathcal{A}} \left( \#steps(s_{\Theta,t} \leftarrow \mathcal{A}(s_{\Theta,t-1})) = s_{\Theta*} \right) \quad (3)$$

Equation 2 finds a set of samples ($s_\Theta$) that maximizes the probability of achieving ideal performance ($f_{ideal}$) for a given code ($\tau$) on the hardware (*exploration*), and Equation 3 encourages finding an algorithm that minimizes search steps by maximizing the reuse of information from previous set of samples ($s_{\Theta,t-1}$) (*exploitation*). In this work, we explore a new possibility using reinforcement learning which strikes a good balance between *exploration* and *exploitation* during the search. In the rest of the paper, we call

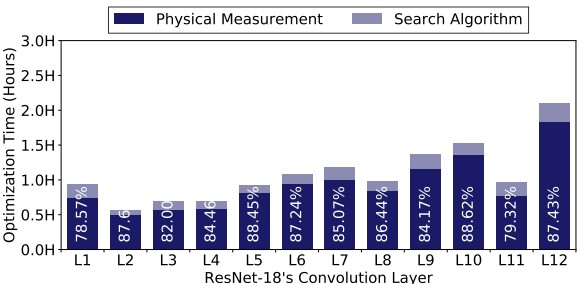

*Figure 2.* AutoTVM (Chen et al., 2018b) optimization time for ResNet-18 (He et al., 2016) on NVIDIA Titan Xp. Numbers in bars denote fraction of time spent on real hardware measurements.

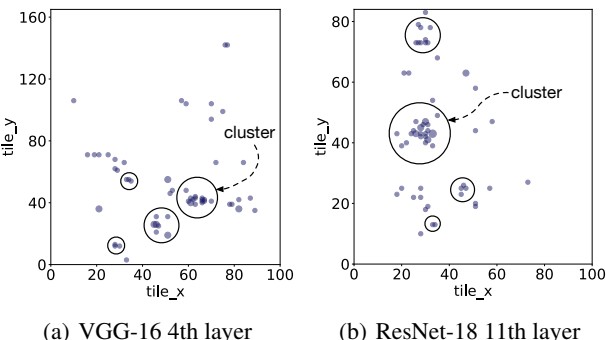

(a) VGG-16 4th layer      (b) ResNet-18 11th layer

*Figure 3.* Illustration of clusters visible among distribution of samples during optimization process in AutoTVM (Chen et al., 2018b)

this the search agent, and we address this in Section 4.1.

**Reducing number of costly hardware measurements.** Figure 2 presents the total and the breakdown of the time it takes to optimize convolution layers of ResNet-18 (He et al., 2016) using AutoTVM (Chen et al., 2018b). It is clear from the graph that majority of the compile time is spent on reaching for measurements on real hardware that is used as a feedback for the aforementioned search algorithms or cost model. Therefore, reducing the frequency of such costly hardware measurements will reduce the overall optimization time significantly. In prior work (Chen et al., 2018b), search algorithms pick fixed number of samples per iteration of optimization and takes a greedy approach in determining which configurations to measure on the real hardware. However, such method overlooks the information from the distribution of the samples while making such decision, which not only leads to longer optimization time from excessive number of hardware measurements but also leaves room for more effective and efficient exploration.

Therefore, the goal of this work is to methodically vary (reduce) the number of configuration samples to measure with regard to the distribution of the samples ($s_\Theta = \{\Theta_1, \Theta_2, ..., \Theta_3\}$) and to intelligently deduce representative points ($s_\Theta' \subset \mathcal{S}_\Theta$) within the design space of configurations that would subsume the subspace ($s_\Theta \subset \mathcal{S}_\Theta$). Therefore, there are two adversarial goals of methodically and intelli-

gently sampling from the distribution to minimize measurements, yet maximize both the potential information ($\mathcal{H}_\Theta$) and the overall fitness ($f$) of configuration samples. Given, $(\mathcal{S}_\Theta, s_\Theta, f, \tau)$, our problem can be formalized into following two conflicting objectives:

$$s'_\Theta = \underset{s_\Theta \subset \mathcal{S}_\Theta}{\operatorname{argmin}} |s_\Theta|$$

$$\text{vs.} \qquad (4)$$

$$s'_\Theta = \underset{s_\Theta \subset \mathcal{S}_\Theta}{\operatorname{argmax}} \big( \sum_{\Theta \in s_\Theta} (\mathcal{H}_\Theta) \cdot \min_{\Theta \in s_\Theta} f(\tau(\Theta)) \big)$$

Figure 3 plots the distribution of sampled configurations by reducing to two dimensions using dimensionality reduction, the observation is that subsets of the sampled configurations are clustered. Since, the variance of the performance among the samples within each cluster is relatively small despite performance differences among different configurations, it is inefficient for the compiler to make measurements on all configurations from each cluster. We leverage this observation and methodically sample representative configurations from the distribution of configurations from the search agent to make our compiler make less hardware measurements without compromising the quality of compilation. We call this sampling module, and address this issue in Section 4.2.

## 4. Reinforcement Learning Compiler with Adaptive Sampling for Efficiency

As discussed in Section 3, there are two distinct yet interrelated issues that have to be addressed for high-performance yet faster compilation. We propose RELEASE[1], reinforcement learning based optimizing compiler with an integrated adaptive sampling to solve this problem. Figure 4 (a) illustrates the framework and its components.

Input to RELEASE are code template ($\tau$), which has information about layers of the input DNN, and the corresponding design space ($\mathcal{S}_\Theta$). RELEASE builds upon prior work's cost model (Chen & Guestrin, 2016) to approximate the design space, and performs search using reinforcement learning based search agent which returns a trajectory ($s_\Theta$). Furthermore, adaptive sampling module adaptively samples from the trajectory ($s'_\Theta$) to minimize number of hardware measurements, which their runtimes are used to determine the best configuration ($\Theta^*$) and used to train the cost model.

In RELEASE, we make two major design choices: (1) we employ reinforcement learning to our search agent for good trade-off regarding *exploration vs exploitation*, and (2) we use clustering based adaptive sampling to minimize hardware measurements without compromising quality of optimization. Rest of the section explains the details of design choices made for each component.

[1]**RELEASE**: **Re**inforcement **Le**arning Compiler with **A**daptive **S**ampling for **E**fficiency

*Table 2.* Hyperparameters used in RELEASE search agent.

| HYPERPARAMETER | VALUE |
| --- | --- |
| Adam Step Size | $1 \times 10^{-3}$ |
| Discount Factor | 0.9 |
| GAE Parameter | 0.99 |
| Number of Epochs | 3 |
| Clipping Parameter | 0.3 |
| Value Coefficient | 1.0 |
| Entropy Coefficient | 0.1 |

### 4.1. Reinforcement Learning based Search Agent

The goal of the search agent is to search for potential configurations. RELEASE makes use of reinforcement learning to ensure that the agent quickly finds the set of good potential configurations. Figure 4 (b) depicts the RELEASE search agent in action. More specifically, RELEASE uses Proximal Policy Optimization (PPO) (Schulman et al., 2017) as its learning algorithm, and Table 2 presents the relevant hyperparameters of the RELEASE search agent.

**State space.** As shown in Table 1, there are several factors that contribute to the performance of the generated code. Each of the knobs, tile_x, tile_y, unroll_explicit, . . . are all different dimensions of optimization. Since these dimensions are interrelated, reinforcement learning based search agent needs to learn about the dependencies among the dimensions of the design space in order to reach optimal overall configuration. We design the state space to contain values for all dimensions of the current configuration.

**Action space.** The agent needs to be able to traverse through the configuration design space. Therefore, we define the action space of the agent as the vector of direction for each dimension of the configuration, and, for every step of the search, our agent aims to take steps towards the optimal configuration. For each and every dimension, the direction is either increment, decrement, or stay.

**Reward formulation.** Reward in RELEASE context is the performance of the output code. However, since the real hardware measurement is very costly in our scenario as discussed in previous sections, we use the estimation from the cost model (Chen & Guestrin, 2016) as a surrogate (or pseudo) reward. As shown in Figure 4 (a), our agent makes queries to the cost model after each episode of search.

**Policy and value networks.** Our search agent uses an actor-critic style policy gradient approach, PPO, which has two networks: policy network and value network. The agent's first layer is shared to foster information sharing among the two networks, and output is fed into the subsequent layers of both networks. Policy network returns vector of directions for each dimension in configuration and value network returns the value of the action.

**Learning procedure.** The whole procedure begins with a set of initial configurations. As shown in Figure 4 (b), for

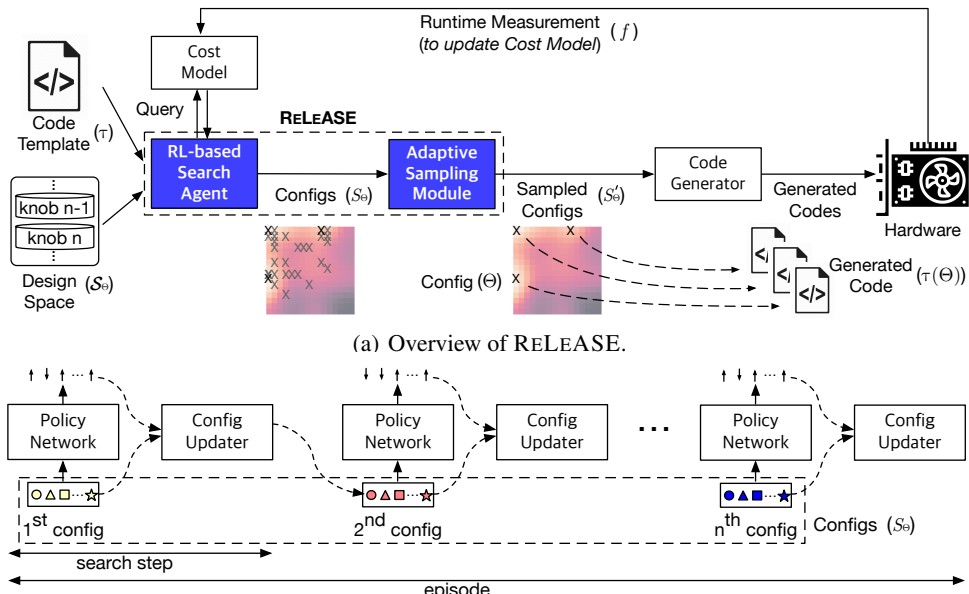

(a) Overview of RELEASE.

(b) RL-based search agent of RELEASE in action.

*Figure 4.* Overview of the RELEASE compilation.

a given input configuration, the agent makes an action, and applying that action to the configuration using configuration updater creates another configuration that would be closer to optimal configuration. Agent takes number of actions in each episode, but in order to avoid unnecessary actions and make the search more efficient, the agent ends the episode after reaching convergence. After each episode, entire trajectory of configurations are evaluated for their fitness by querying to the cost model. Agent then formulates the return values of the cost model as reward and trains the policy and value networks, which help the agent learn about the design space. By repeating this process, the agent gradually learns to understand the interplay between different dimensions on the input in order to locate good configurations. After repeating several episodes, the agent feeds trajectory of configurations ($s_\Theta$) into our adaptive sampling module.

### 4.2. Adaptive Sampling Module

From Section 3.2, we notice that physical hardware measurements are costly and take up majority of the optimization time. Number of hardware measurements is a major contributor to prolonging the optimization time, and methodical way of reducing the measurements will reduce the optimization time significantly. We propose a clustering based sampling algorithm that adaptively samples configurations from the input trajectory to reduce the number of hardware measurements yet maintain or even augment the quality of the samples to be sent to real hardware, improving both the effectiveness and the efficiency of the overall compiler

**Adaptive sampling algorithm.** We illustrate our adaptive sampling algorithm in Algorithm 1. By taking advantage of the observation from Section 3.2, the algorithm starts by clustering the samples of the input search trajectory. We use

---

**Algorithm 1** Adaptive Sampling Algorithm

1: // $s_\Theta$: search trajectory, $v_\Theta$: visited configurations
2: **procedure** ADAPTIVESAMPLING($s_\Theta, v_\Theta$)
3:     $NextSamples = \emptyset, PreviousLoss = \infty$
4:     **for** $k$ **in** range(8, 64) **do**
5:         $Centroids, Clusters, Loss = k\text{-means}(s_\Theta, k)$
6:         // exit loop at *knee* of loss curve
7:         **if** $Constant \times Loss > PreviousLoss$ **then**
8:             break
9:         **end if**
10:         $PreviousLoss = Loss$
11:     **end for**
12:     $NextSamples = Centroids$
13:     // replace visited configuration with new ones
14:     **for** $c$ **in** $Centroids$ **do**
15:         **if** $c$ **in** $v_\Theta$ **then**
16:             $NextSamples.\text{replace}(c, \text{mode}(s_\Theta))$
17:         **end if**
18:     **end for**
19:     // make measurements on hardware
20:     **return** $NextSamples$
21: **end procedure**

---

$k$-means clustering to determine centroids of configurations, because $k$-means clustering been shown to be effective in finding clusters and because it only requires $k$ to be determined over $\epsilon$ or $radius$ in other clustering algorithms like DBSCAN (Ester et al.) or mean-shift clustering (Comaniciu & Meer, 2002), which need to be determined relative to the dimensions of the search space making it more difficult than a fixed value, $k$. Determining the number of clusters, $k$, is a hyperparameter that is ambiguous and entails recognizing the trade-off between the gains from reducing number of

clusters and the downside of increased loss from the reduction. In the context of optimizing compiler, reduced $k$ leads to shorter optimization time while increased loss that comes from the reduction leads to loss of underlying information from the input search trajectory. Our algorithm iterates through various $k$ until it hits the *knee* of the loss curve of the $k$-means algorithm: optimal trade-off point between more physical measurements and faster optimization.

After the clustering process, subset of the centroids may be redundant with the previously visited configurations. Therefore, the our sampling algorithm checks the history ($v_\Theta$) to sift out previously visited configurations from the centroids, and replaces them with configuration generated from modes of each dimension. This process not only removes redundancy but also increases the potentially meaningful exploration that maximizes the information ($\mathcal{H}_\Theta$) of the sampled configurations. Finally, sampled configurations ($s'_\Theta$) are passed onto code generator to be run on hardware and the resulting runtimes are used to update the cost model.

## 5. Evaluation

We integrate RELEASE optimizing compiler into TVM (Chen et al., 2018a) to perform component evaluation of RELEASE and compare with AutoTVM (Chen et al., 2018b). We first evaluate components of RELEASE in Section 5.1 and Section 5.2 on set of convolution layers sampled from AlexNet (Krizhevsky et al., 2012), VGG-16 (Simonyan & Zisserman, 2014), and ResNet-18 (He et al., 2016). Then we evaluation of RELEASE on both set of layers and end-to-end deep models, in Section 5.3.

### 5.1. Reinforcement Learning based Search Agent: Improving Efficacy of Search Algorithm

In the previous approach (Chen et al., 2018b), authors have used simulated annealing to find potentially optimal configurations on top of the fitness estimation from the cost model. Figure 5 compares the number of search steps taken per iteration to reach or converge to the solution in simulated annealing and reinforcement learning, respectively. Overall, observation is that RELEASE's reinforcement learning

Table 3. Details of the DNN models used in evaluating RELEASE.

| NETWORK | DATASET | NUMBER OF TASKS |
|---|---|---|
| AlexNet | ImageNet | 5 |
| VGG-16 | ImageNet | 9 |
| ResNet-18 | ImageNet | 12 |

Table 4. Details of the layers used in evaluating RELEASE.

| NAME | MODEL | LAYER TYPE | TASK INDEX |
|---|---|---|---|
| L1 | AlexNet | convolution | 1 |
| L2 | AlexNet | convolution | 4 |
| L3 | VGG-16 | convolution | 1 |
| L4 | VGG-16 | convolution | 2 |
| L5 | VGG-16 | convolution | 4 |
| L6 | ResNet-18 | convolution | 6 |
| L7 | ResNet-18 | convolution | 9 |
| L8 | ResNet-18 | convolution | 11 |

agent requires 2.88×less search steps compared to simulated annealing to find good solution. This comes from reinforcement learning agent's ability to (1) quickly learn about the correlation between different dimensions, and (2) start search on top of previous iterations, to reuse the information, over starting from scratch, relying on stochastic guarantees of the simulated annealing process.

### 5.2. Adaptive Sampling Module: Reducing Number of Costly Hardware Measurements

Figure 6 summarizes the effect of applying RELEASE's adaptive sampling module on simulated annealing and reinforcement learning search. First, results show that using adaptive sampling helps the framework make less hardware measurements regardless of the search algorithm. The adaptive sampling algorithm reduces the number of measurements by 1.98×when used with simulated annealing and 2.33×with reinforcement learning. One observation is that the adaptive sampling is more effective with reinforcement learning. This comes from the reinforcement learning agent's capacity to better localize the search to meaningful samples (*exploitation*) while still finding good solution by maintaining diversity (*exploration*). Next, we will confirm that these reductions do not hurt optimization performance.

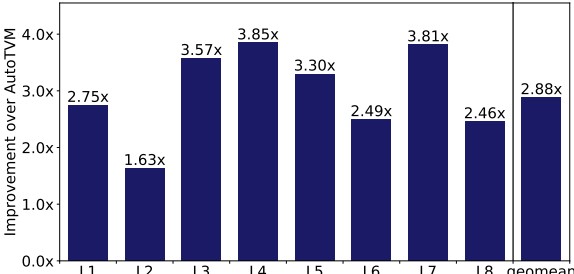

Figure 5. Reduction in number of steps for convergence.

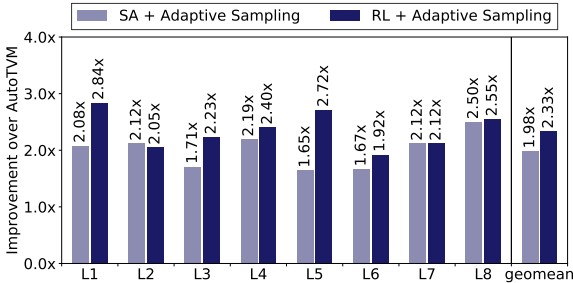

Figure 6. Reduction in number of hardware measurements.

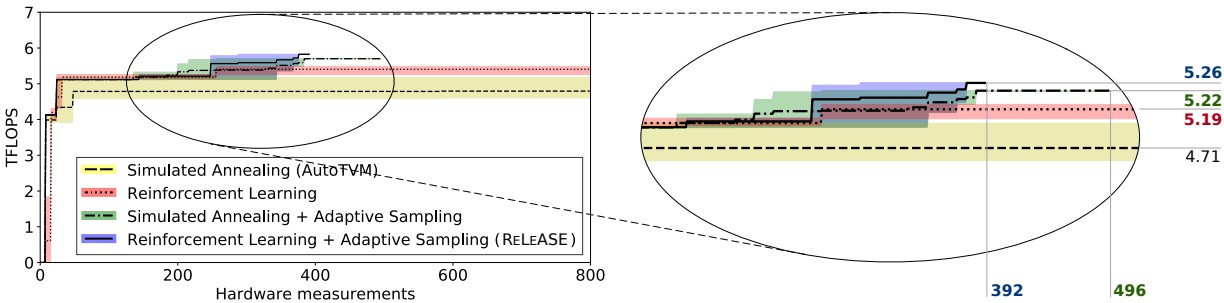

*Figure 7.* Layer evaluation of output performance for ResNet-18 (He et al., 2016) 11th layer.

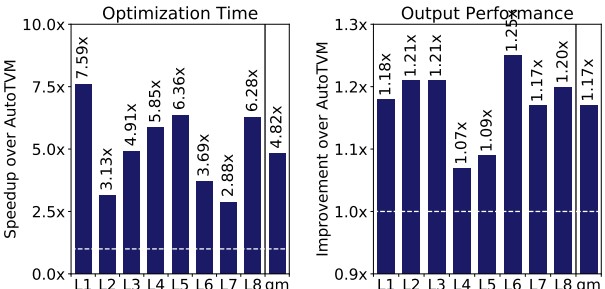

*Figure 8.* Layer and end-to-end evaluation. Dashed lines denote AutoTVM (Chen et al., 2018b) performance.

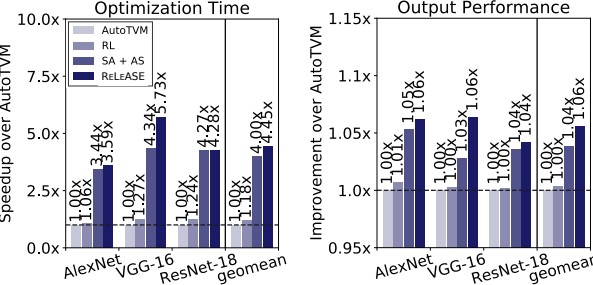

*Figure 9.* Layer and end-to-end evaluation. Dashed lines denote AutoTVM (Chen et al., 2018b) performance.

### 5.3. Putting It All Together: Reducing Optimization Time & Output Inference Time

RELEASE integrates two components into the workflow: reinforcement learning based search agent and adaptive sampling module. This section compare the performance of the integrated RELEASE with AutoTVM (Chen et al., 2018b) on both set of layers and end-to-end deep networks, presented in Table 4 and Table 3.

**Layer evaluation.** Figure 7 shows the trend of output code performance of ResNet-18's 11th layer over number of hardware measurements during optimization. The figure illustrates that the reinforcement learning search finds better configurations than simulated annealing which results in better output code performance, and the adaptive sampling reduces number of hardware measurements significantly during optimization. Also, RELEASE's reinforcement learning search and adaptive sampling working in tandem emits better code with shorter optimization time than others.

As such, Figure 8 compares optimization time and the performance of the output code in RELEASE and AutoTVM to confirm the observation. RELEASE achieved 1.17×better performance with 4.82×shorter optimization time compared to AutoTVM. Overall, the results suggest that the reinforcement learning based search agent makes effective search over the design space, and adaptive sampling module reduces hardware measurements and overall optimization time while even improving output performance.

**End-to-end evaluation.** Up until now, we have focused on evaluation with subset of layers. Now we continue our discussion to the applicability of RELEASE to optimization of end-to-end deep neural networks. Figure 9 shows that RE-LEASE spends 3.59×, 5.73×, and 4.28×less time than AutoTVM to optimize AlexNet, VGG-16, and ResNet-18, respectively. On average, our work shows 4.45×optimization time speedup while achieving up to 6.4%improvement in terms of performance of output code. Inference time in Figure 9 illustrates the speedup for optimized code. Raw numbers are available in Table 5 and Table 6. All in all, such improvements result from more efficient search algorithm and the reduced number of hardware measurements from adaptive sampling algorithm.

## 6. Related Works

RELEASE uniquely offers a solution that exclusively enables (i) reinforcement learning and (ii) efficient sampling in the context of (iii) optimizing compilers for neural networks. As such, we discuss the related work from each of the three independent research directions.

**Optimizing compilers.** TensorComprehensions (Vasilache et al., 2018) and TVM (Chen et al., 2018a) use genetic algorithm and simulated annealing to choose parameters of polyhedral optimization for neural networks. In a more general context, some computing libraries (Whaley & Dongarra, 1998; Frigo & Johnson, 1998) make use of black box optimization and also profiling-based com-

*Table 5.* Raw numbers of optimization time for end-to-end evaluation.

| NETWORK | AutoTVM | RL | SA + AS | RELEASE |
|---|---|---|---|---|
| AlexNet (Krizhevsky et al., 2012) | 4.31 Hours | 4.06 Hours | 1.25 Hours | **1.20 Hours** |
| VGG-16 (Simonyan & Zisserman, 2014) | 11.2 Hours | 8.82 Hours | 2.57 Hours | **1.95 Hours** |
| ResNet-18 (He et al., 2016) | 9.13 Hours | 7.39 Hours | 2.14 Hours | **2.13 Hours** |

*Table 6.* Raw numbers of output performance for end-to-end evaluation.

| NETWORK | AutoTVM | RL | SA + AS | RELEASE |
|---|---|---|---|---|
| AlexNet (Krizhevsky et al., 2012) | 1.0277 ms | 1.0207 ms | 0.9762 ms | **0.9673 ms** |
| VGG-16 (Simonyan & Zisserman, 2014) | 3.9829 ms | 3.9710 ms | 3.8733 ms | **3.8458 ms** |
| ResNet-18 (He et al., 2016) | 1.0258 ms | 0.9897 ms | 0.9897 ms | **0.9831 ms** |

pilation passes (Chang et al., 1991; Novillo, 2014) utilize runtime information to generate optimized code. Later, AutoTVM (Chen et al., 2018b) incorporates learning with boosted trees within the cost model for TVM to reduce the number of real hardware measurements. While RELEASE in inspired and builds on these prior work, unlike them, it is based on reinforcement learning and further reduces the number of measurements by focusing them through adaptive sampling that leverages clustering.

**Reinforcement learning for hyperparameter optimization.** There are a growing body of studies on using reinforcement learning to perform various optimizations (Gao et al., 2018; Mirhoseini et al., 2017; Mao et al., 2016; Ye & Li, 2018; Henderson et al., 2017; Dong et al., 2018; Xu et al., 2018; Jaderberg et al., 2017) for a variety of objectives including hyperparameter optimization for neural networks. For instance, DeepArchitect (Negrinho & Gordon, 2017) and NAS (Zoph & Le, 2017; Pham et al., 2018) use reinforcement learning to automate the process of designing deep neural network models and their associated parameters. HAQ (Wang et al., 2018) and ReLeQ (Elthakeb et al., 2018) use reinforcement learning to chose levels of quantization for the layers of a given deep neural network. AMC (He et al., 2018) formulates neural network compression as a RL problem. Our work exclusively explores a different problem, that is optimizing compilers, using reinforcement learning.

**Sampling algorithms for learning.** Active learning is a broad field (Settles, 2009; Sugiyama, 2006; Cai et al., 2013; Wu et al., 2019; Chen & Price, 2017; Goetz et al., 2018; Dasgupta & Hsu, 2008; Huang et al., 2010; Beygelzimer et al., 2008) that uses a measure of change in the model to decide which training data elements should be used to update the model. Passive learning (Yu & Kim, 2010; O'Neill et al., 2017) is an alternative view that, independent of the model, analyze the distribution of the training data set and selects a subset. The adaptive sampling algorithm for RELEASE shares similarities with Passive learning but it differs in its context. The sampling is designed to reduce the number of samples from the trajectory of search whilst performing an optimization to accelerate the process.

## 7. Conclusion

This paper is an initial effort to bring reinforcement learning to the realm of optimizing compilers for neural networks. While devising an RL-based optimizing compiler, called RELEASE, we also developed an adaptive sampling algorithm to reduce the samples required to navigate the search space. Experimentation with real-world deep models shows that RELEASE not only reduces the time for compilation significantly, but also improves the quality of the code. This encouraging result suggests a significant potential for reinforcement learning to optimizing deep learning models.

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
