# OpenReview forum: "Reinforcement Learning and Adaptive Sampling for Optimized DNN Compilation"
_ICML.cc/2019/Workshop/RL4RealLife — RL4RealLife 2019_

### Official Review · AnonReviewer1 · 2019-05-24
**Accept. Well written and evaluated. But not a reinforcement learning problem. Bandit.**

**Rating:** 4
**Confidence:** 4

**Review:**

This paper proposes a reinforcement learning algorithm for compiler optimization. The state is a vector of tunable knobs of an optimized compiler. The reward is a pseudo-reward from a simulator of how the compiler performs. The algorithm works as follows. In round t, it selects a trajectory in the space of tunable knobs using a policy that is trained on the simulator. The simulator is trained on the rewards in the first t - 1 rounds. The trajectory of the executed policy is reduced to representative states; and the rewards in those states are measured and used to update the simulator.

This is a nice paper and the approach is comprehensively evaluated.

My main comment is that this is not a reinforcement learning problem. In particular, note that the learning agent has a complete control over the setting of the compiler. Therefore, the agent does not have to plan for moving in the space of tunable knobs. Therefore, your problem is bandit problem, where the feature vector is a vector of tunable knobs and you optimize a potentially complex function of the knobs. To the best of my knowledge, the best approach for this class of problems is a Gaussian process bandit

  https://arxiv.org/abs/0912.3995

---

### Official Review · AnonReviewer2 · 2019-05-25
**sensible approach to RL for compilers, might benefit from more polishing.**

**Rating:** 4
**Confidence:** 2

**Review:**

The authors describe an RL approach to speed up neural network compilation. Given that compilation is a search problem in a very high dimensional search space, this is a reasonable approach.

This reviewer found the adaptive sampling method not so well motivated, it would make sense to position this more in the model-based RL context, where a surrogate model of the environment is used so the real environment does not have to be expensively interacted with too often.


Positive:
It is without question that getter faster DNN code is important ☺

Negative:
Quite specific agent designed for only one compilation task, which is, to be fair, listed as a limitation in the paper.

Disclaimer: This reviewer is not a computer scientist, and has therefore a very limited understanding of how compilers work.

Suggestion: Overall, the paper would benefit from additional proofreading with a focus on grammar and clarity.

---

### Decision · Program_Chairs · 2019-05-28

Accept